# The Effect of Laser Micro Grooved Platform Switched Implants and Abutments on Early Crestal Bone Levels and Peri-Implant Soft Tissues Post 1 Year Loading among Diabetic Patients—A Controlled Clinical Trial

**DOI:** 10.3390/medicina58101456

**Published:** 2022-10-15

**Authors:** Lalli Dharmarajan, P. S. G. Prakash, Devapriya Appukuttan, Jasmine Crena, Sangeetha Subramanian, Khalid J. Alzahrani, Khalaf F. Alsharif, Ibrahim F. Halawani, Mrim M. Alnfiai, Ahmed Alamoudi, Mona Awad Kamil, Thodur Madapusi Balaji, Shankargouda Patil

**Affiliations:** 1Department of Periodontics, SRM Dental College and Hospital, Ramapuram, Chennai 600089, India; 2Department of Clinical Laboratories Sciences, College of Applied Medical Sciences, Taif University, P.O. Box 11099, Taif 21944, Saudi Arabia; 3Department of Information Technology, College of Computers and Information Technology, Taif University, P.O. Box 11099, Taif 21944, Saudi Arabia; 4Oral Biology Department, Faculty of Dentistry, King Abdulaziz University, Jeddah 21589, Saudi Arabia; 5Department of Preventive Dental Sciences, College of Dentistry, Jazan University, Jazan 45412, Saudi Arabia; 6Department of Dentistry, Tagore Dental College and Hospital, Chennai 600127, India; 7College of Dental Medicine, Roseman University of Health Science, South Jordan, UT 84095, USA

**Keywords:** diabetes mellitus, laser micro-grooved implants, laser micro-grooved abutments, mean crestal bone level measurements, relative position of the gingival margin

## Abstract

*Background and Objectives:* The study aimed to compare the mean crestal bone level (CBL) and peri-implant soft tissue parameters in laser micro-grooved (LMG) platform switched implants and abutments (I&A) post 1 year of functional loading among non-diabetic and type II diabetic individuals. *Materials and methods:* Patients with an edentulous site having minimum bone height and width of ≥13 mm and ≥6 mm, respectively, were divided into two groups: (i) Non-diabetic-8 (control) and (ii) diabetic-8 (test). LMG Implants were placed and loaded immediately with a provisional prosthesis. Mean crestal bone level (MCBL) was evaluated radiographically at baseline and at 1 year. Peri-implant attachment level (PIAL) and relative position of the gingival margin (R-PGM) were recorded. Implant stability quotient (ISQ) level and implant survival rate (ISR) were evaluated at 1 year. *Results:* Early MCBL within the groups 1 year postloading was similar both mesially and distally (control—0.00 to 0.16 mm and 0.00 to 0.17 mm, respectively; test—0.00 to 0.21 mm and 0.00 to 0.22 mm, respectively) with statistical significance (*p* ≤ 0.003, *p* ≤ 0.001 and *p* ≤ 0.001, *p* ≤ 0.001, respectively). However, intergroup comparison showed no significant difference statistically in the MCBL in 1 year post functional loading. The peri-implant soft tissue parameters showed no significant difference between the groups. ISQ level between both groups did not reveal any significant changes (*p* ≤ 0.92), and ISR was 100%. *Conclusions:* LMG Implants resulted in minimal and comparable early crestal bone loss and soft tissue changes post 1 year of functional loading in moderately controlled diabetic and non-diabetic individuals, suggesting that this could be a reliable system for use in systemically compromised individuals.

## 1. Introduction

The survival of dental implants is determined by factors such as implant design, surface properties, and surgical protocol [1]. Initially successful osseointegration determines implant stability, and later the bone remodeling associated with prosthetic loading and crown placement determines the implant survival rate (ISR). In addition, patient-related local and systemic conditions such as diabetes mellitus are vital modifiers of implant survival and success rates [2].

Chronic hyperglycemia is linked to poor wound healing, altered bone metabolism, and hyperinflammatory responses [3]. In the presence of elevated glycemic levels, there is decreased collagen production during callus formation, resulting in apoptosis of bone lining cells and increased osteoclastic activity, which interferes with osseointegration [4]. In addition, autoimmune reactions are induced in tissues with increased osteoclastic activity, resulting in bone resorption and associated interference with osteoblastic activity. The sensitivity of the parathyroid glands is altered, resulting in an imbalance in calcium and phosphorus homeostasis, which has a negative impact on cellular functions and the extracellular matrix of the bone [5]. Thus, the glycemic status affects the implant survival rate by altering the bone metabolism throughout the healing phase. Therefore, there is a need to understand and address the significance of this potential risk factor in dental implants-related research [6].

Studies have shown that the implant stability parameters, marginal bone loss around implants, soft tissue inflammation, and bleeding on probing increased in proportion to increasing HbA1c levels [7]. Oates et al. reported that moderately controlled diabetics demonstrate a significant difference in mean crestal bone loss but with better implant survival rate [8]. Multiple studies have evidenced that individuals with moderately and poorly controlled type II diabetes have an excellent overall implant survival rate with greater crestal bone loss when compared to non-diabetic individuals. With better implant survival rate in diabetics, optimizing implant designs aimed at reducing crestal bone loss could significantly improve their quality of life [8,9,10]. Very limited studies have evaluated the potential effect of implant designs on reducing crestal bone loss in diabetic individuals and specifically on moderately controlled diabetic individuals, and this needs further exploration.

Implant design modifications aim at reducing crestal bone loss and achieving better osseointegration, and recently laser-ablated micro-channels or micro-grooves placed within the implant collar have been shown to limit apical migration of the junctional epithelium and prevent crestal bone loss [11]. Histological studies have further suggested that a laser micro-grooved (LMG) implant surface allowed direct contact of a stable connective tissue with an intact biological seal, giving a cold welding effect to the implant collar, and a more robust perpendicular collagen fiber attachment preventing epithelial down growth and crestal bone resorption post 1 year of loading [12,13,14]. In a recently conducted microbiological and longitudinal study, reduced microbial load was observed at 18 months with minimal crestal bone loss at 3 years in healthy individuals. These results were attributed to laser micro-grooving along with platform switching characteristics [15,16]. Thus, the studies have emphasized that the LMG implant design modifications provided an effective result in the reduction of crestal bone loss in normal healthy individuals. Therefore, the special characteristics, features, and one-year follow-up capabilities of LMG implants when compared to non-diabetic and well-controlled individuals could be used as an advantage to prevent crestal bone loss in diabetic individuals. 

No prior studies have evidenced the use of LMG platform switched implants and abutments in preventing crestal bone loss in moderately controlled diabetic patients. Considering the advantages of LMG platform switched implants and abutments, this study hypothesized that the abovementioned precisely designed implants and abutments could have a positive influence on minimizing the mean crestal bone loss and peri-implant attachment loss in moderately controlled diabetic individuals at 1 year post functional loading comparable to that of non-diabetic individuals.

## 2. Materials and Methods

### 2.1. Trial Design

This prospective clinical trial was based on a cohort of patients seeking an implant-supported restoration at the Department of Periodontology, SRM Dental College, Ramapuram, Chennai-89 between January 2020 to October 2021. The study was declared to the Institutional Scientific Committee and Ethical Review Board and prior approval was obtained (IRB APPROVAL no. SRMDC/IRB/2019/MDS/No.503). The clinical trial registry number is as follows: REF/2019/12/030040[DE].

### 2.2. Participants

The participants of the present study were included based on the study group allocation (Figure 1), and were either systemically healthy (control group) or type II diabetic (test group). The general inclusion criteria included male or female patients aged ≥30 to 60 years, with mandibular premolars and molar edentulous sites with sufficient bone height (at least 13 mm) and sufficient bone width (at least 6 mm). The test group, which included diabetic patients with HbA1c levels of 8.1 to 10 (moderately controlled diabetic individuals) was recruited for the study. The study sample was calculated based on results obtained from a study by Aguilar-Salvatierra et al. in 2016 [17]. Taking a 20% dropout rate into consideration, the sample size was increased to 20 edentulous sites that require implant placements with 10 (LMG) platform switched implants and abutments to be placed in each group. 

### 2.3. Outcome Measures

The primary objective of this study was to evaluate and compare the MCBL changes radiographically (mesially and distally) at baseline (immediately after restoration) and 1 year post functional loading in both non-diabetic (control group) and diabetic (test group) sites, and additionally also to assess and compare the implant survival rates 1 year post functional loading between both non-diabetic (control group) and diabetic (test group) sites. 

The secondary objective of this study was to evaluate and compare the relative position of gingival margin (R-PGM) on LMG platform switched implants with LMG abutments among non-diabetic (control group) and diabetic (test group) patients at baseline, 6 months, and 1 year post functional loading. Additionally, we aimed to assess the implant stability quotient (ISQ) values using resonance frequency analysis and compare it between non-diabetic (control group) and diabetic (test group) patients prior to prosthetic restoration.

Hypotheses: considering the advantages of laser micro-grooved platform switched implants and abutments, this study hypothesized that the abovementioned precisely designed implants and abutments could have a positive influence on minimizing the mean crestal bone loss and peri-implant attachment loss in moderately controlled diabetic individuals 1 year post functional loading comparable to that of non-diabetic individuals.

#### 2.3.1. Clinical Parameters to Be Evaluated Include

Full mouth plaque scores (FMPS) [15]: evaluated at baseline (before implant placement) and 1 year post functional loading.

Full mouth Bleeding scores (FMBS) [15]: evaluated at baseline (before implant placement) and 1 year post functional loading.

Periodontal probing depth (PPD) [15]: evaluated at baseline (before implant placement) and 1 year post functional loading.

Clinical attachment level [16]: evaluated at baseline (before implant placement) and 1 year post functional loading.

#### 2.3.2. Site-Specific Parameters to Be Evaluated

Site-specific plaque scores (S-SPS) [15]: evaluated immediately post restoration (baseline), 6 months, and 1 year post functional loading.

Site-specific bleeding scores (S-SBS) [16]: evaluated immediately post restoration (baseline), 6 months, and 1 year post functional loading.

Peri—Implant sulcus depth (PISD) [16]: evaluated immediately post restoration (baseline), 6 months, and 1 year post functional loading.

Peri—Implant abutment attachment level (PIAL) [15]: evaluated immediately post restoration (baseline), 6 months, and 1 year post functional loading.

Relative position of gingival margin (R-PGM) [16]: evaluated immediately post restoration (baseline), 6 months, and 1 year post functional loading.

ISQ [17,18,19]: evaluated immediately after implant placement for restoration.

Radiographic parameters to be evaluated using RVG [16]:

MCBL changes of mesial and distal aspect (M-MCBL and D-MCBL): evaluated immediately post restoration (baseline) and 1 year post functional loading.

ISR [6]: evaluated 1 year post functional loading. 

### 2.4. Randomization

Patients who fulfilled the inclusion and exclusion criteria were enrolled for this controlled clinical trial. In order to rule out examiner bias, a single calibrated examiner performed all of the clinical parameters. Routine preliminary phase assessments and treatment were performed. Patients who had fulfilled the abovementioned inclusion and exclusion criteria, and who were willing to participate in the study, were selected and divided into two groups. A total of 20 edentulous sites requiring implant placement were recruited for this study. All of the surgical procedures were performed by a single well-experienced operator. The radiographic parameters were assessed by a separate calibrated examiner who was unaware of the recruitment and the clinical parameters assessed.

### 2.5. Interventions

A single calibrated operator who was blinded about the diabetic or non-diabetic sites performed the surgical procedures. After administration of local anesthetics (2% lignocaine with 1:80,000 adrenaline) patients were instructed to rinse with 0.12% chlorhexidine solution for 30 s. The implant sites were prepared under local anesthesia, and a mid-crestal incision followed by minimal flap elevation and a pilot drill were driven initially with 950 rpms and 35 Ncm. An implant surgical kit was used to prepare the osteotomy sites and place the implants. Once the final osteotomy was carried out, the implants were driven with an insertion torque of more than 35 Ncm using the implant drive (Figure 2). ISQ was measured using resonance frequency analysis (RFA) (Penguin^RFA^ unit, Integration Diagnostics, Goteborg, Sweden), performed by placing a transducer on the implant collar and the reading of 60 (osseointergration diagnostics) to confirm the primary stability [17,18,19]. When the reading of the ISQ limit touched 60 and above, it indicated immediate loading. All of the implants were placed equi-crestally. This was followed by the placement of a prosthetic abutment and was restored immediately with functional loading (Figure 3). The same surgical protocol was followed for both the control and test groups. All patients received postoperative instructions to rinse with 0.2% chlorhexidine digluconate twice daily for a two-week period. Analgesic medication (ibuprofen, 500 mg) was prescribed.

### 2.6. Statistical Analysis

The collected data were analyzed with IBM SPSS Statistics for Windows, Version 23.0. (Armonk, NY: IBM Corp). To find the significant difference between the bivariate samples among the paired groups (baseline and 1 year), a Wilcoxon signed rank test was used. To find the significant difference between the bivariate samples among independent groups (baseline and 1 year) a Mann–Whitney U test was used. To compare the continuous variables between two groups, i.e., intergroup comparison of FMPS, FMBS, PPD, CAL, S-SPS, S-SBS, PISD, PIAL, RPGM, MCBL, ISR, and ISQ, Student’s t-test was used.

Intragroup comparison of all of the clinical and radiographic parameters within the control and the test groups were carried out using a paired t-test. In both the above statistical tools the probability value 0.05 was considered a significant level. 

## 3. Results

The mean age of the participants was 38.62 ± 5.06 years and 42.62 ± 5.34 years in the control and the test groups, respectively. Gender distribution in the control and the test groups was 5 male /5 female and 4 male/6 female participants, respectively. The groups did not differ based on age and gender distribution (*p* = 0.46 and *p* = 0.17, respectively). 

Table 1: FMPS and FMBS showed a statistically significant reduction from 15% and 11% at baseline to 10% and 8% at 1 year post functional loading in the control and the test groups, respectively (*p* ≤ 0.001). Intergroup comparison of FMPS and FMBS showed no significant difference between the groups at both time points (*p* = 0.52). PPD in both the control and the test groups showed a minimal reduction from 2.94 to 2.90 mm and from 2.88 to 2.63 mm, respectively, from baseline to 1 year post functional loading with no significant difference (*p* = 0.90). No attachment loss was observed in both the control and the test groups throughout the study period. Intergroup comparison of PPD between the control and test groups at both time points showed no statistical significance (*p* = 0.43 and *p* = 0.39, respectively). 

Table 2: Intragroup comparison of SS-PS in both the control and test groups showed a similar reduction in plaque deposition, i.e., 9% at baseline, 8% at 6 months, and 6% at 1 year post functional loading. There was a statistically significant reduction in the plaque score from baseline to 1 year post functional loading, i.e., 9% to 6%, *p* ≤ 0.05, in both groups (Table 3). A similar reduction in the SS-BS was observed in the control and the test groups from baseline to 1 year post functional loading, i.e., 9%, 8%, and 6%, and 9%, 7%, and 6%, respectively. A significant reduction was observed at 1 year from baseline in both the control and the test groups, *p* ≤ 0.05 (Table 3).

Table 3: Intergroup comparison of SS-PS between the control and test groups revealed almost similar plaque scores at all time intervals, i.e., baseline (9% vs. 9%), 6 months (8% vs. 8%), and 1 year (6% vs. 6%) with no statistical significance. Likewise, the SS-BS between the control and test groups revealed almost the same bleeding scores at all time intervals, i.e., baseline (9% vs. 9%), 6 months (8% vs. 7%), and 1 year post functional loading (6% vs. 6%) with no statistical significance. 

Table 4: Intragroup assessment of PISD in the control and test groups revealed minimal reduction in the sulcus depth from baseline to 1 year post loading with no significant changes at any time points. Intergroup comparison of the PISD between the control and the test groups at baseline, 6 months, and 1 year post functional loading revealed almost similar probing depths at all time intervals, i.e., baseline (2.11 vs. 2.03 mm), 6 months (2.09 vs. 2.04 mm) and 1 year post functional loading (2.06 vs. 2.07 mm) with no significant difference between the groups. No peri-implant attachment loss was observed in both groups throughout the study period.

The R-PGM in the control and test groups revealed no changes in relation to the restored implant crown at any of the evaluated time points (3.37 ± 0.20 mm and 3.53 ± 0.30 mm, respectively). Furthermore, comparison of the R-PGM between the control and test groups at all three timepoints revealed no changes in the position of the gingival margin, i.e., 3.37 vs. 3.53 mm, respectively, with no difference between the groups statistically (*p* = 1). 

Intragroup comparison of MCBL in the control group in both the mesial and distal aspects showed radiographically minimal early crestal bone loss from baseline to 1 year post functional loading, i.e., 0.00 to 0.16 mm in the mesial aspect and 0.00 to 0.17 mm in the distal aspect with statistical significance (*p* ≤ 0.003 and *p* ≤ 0.001, respectively). Similarly, radiographically minimal early crestal bone loss was observed in the test group, i.e., 0.00 to 0.21 mm in the mesial aspect and 0.00 to 0.22 mm in the distal aspect from baseline to 1 year post loading with a significant difference statistically (*p* ≤ 0.001 and *p* ≤ 0.001, respectively); see Table 4. No statistically significant difference was observed between both groups in the M-MCBL and D-MCBL at 1 year post functional loading (Figure 4).

ISR between the control and test groups after 1 year revealed 100% survival of all of the implants with no failures in both groups. ISQ between the control and test groups showed similar implant stability quotients of 74.50 ISQ (control group) vs. 74.25 ISQ (test group), which was greater than the permissible limits, indicative that all implants were ready for immediate loading protocol. The ISQ levels between the groups did not reveal any significant changes (*p* = 0.92).

## 4. Discussion

Titanium alloys have superior mechanical properties, corrosion resistance, and biocompatibility, favoring their wide use in the manufacturing of medical devices. The bioactivity of these alloys is attributed to the presence of a dense and coherent film of a nanometric thick passivation layer composed of TiO2 [20,21]. Newer metal ingots have been introduced in the field of implant dentistry; however, they are at a more novice stage requiring more research before they can be incorporated into clinical practice. Extensive research aiming at micro-structural modification of dental implants such as titanium nanotubes and laser-based technologies have been shown to improve peri-implant parameters and consequently implant survival rate [22,23].

Diabetes mellitus was previously recognized as a relative risk factor for dental implants’ survival [24]. However, currently, there is a change in the paradigm and studies indicate that diabetes patients benefit from oral rehabilitation with dental implants [10]. In this study it was hypothesized that in moderately controlled diabetics prone to compromised peri-implant soft and hard tissue healing, surface-modified implant design with Laser-Lok technology could be beneficial. This design allows cells such as osteoblasts and fibroblasts to link and organize themselves optimally in the laser micro-channels, creating a biological seal along the abutment per se and osseointegration along the implant collar with a cold welding effect, which collectively contributes to minimal inflammatory infiltration at the crest module of the implant along with a soft tissue seal, thereby minimizing microbial colonization, contributing to better implant stability, maintaining peri-implant health, and minimizing early crestal bone loss [25,26,27]. 

Katyayan et al. suggested that despite the metabolic disparities evident in diabetic patients, an early loading regimen might be successful. Likewise, Al Amri et al. [28] reported identical clinical and radiographic outcomes in terms of soft tissue conditions, crestal bone levels, and implant success rates in type II diabetic and non-diabetic patients based on the early loading strategy. Therefore, in this study immediate implant loading in both groups was chosen, as it could not only enhance successful function but also help in better metabolic outcomes.

The PPD in both the control and the test groups reduced evenly and remained at comparable levels at 1 year post functional loading with no significant difference. Likewise, no CAL was observed in both groups. The results indicate that the patients did not experience any active periodontal disease during the study period. The S-SPS on the abutment surfaces in both the control and the test groups were almost similar from baseline to 6 months and from 6 months to 1 year post functional loading with no statistical difference. The plaque accumulation on the abutment surface significantly reduced from baseline to 1 year in both groups with statistical significance. The similar reduction in plaque accumulation in both the control and the test groups could be attributed to the similar implant and abutment design. 

Likewise, almost similar bleeding scores were observed from baseline to 6 months and from 6 months to 1 year post functional loading with no statistical significance in both groups. The bleeding scores significantly reduced from baseline to 1 year in both groups with statistical significance. Intergroup comparison revealed similar bleeding scores at all three time intervals, with no difference between the groups statistically. The similar percentages of S-SPS and S-SBS indicate that LMG on the implants surface results in an early connective tissue attachment seal, allowing for a good mucosal barrier in the collar region, thereby reducing plaque accumulation and reducing gingival inflammation. 

Intra- and intergroup assessments of PISD in the control and the test groups revealed minimal reduction or almost similar probing depth measurements with no significant difference between the time points. No peri-implant attachment loss was observed in both the control and test groups. These findings further underscore the previous findings that Laser-Lok implants with LMG abutments established a better mucosal barrier in the collar area around the implant–abutment connection, which was almost on par with the soft tissue gingival fiber attachment in the natural tooth, creating a dense soft tissue barrier [29,30]. The resistance to probing attributed to the perpendicular orientation of collagen fibers in the connective tissue to the abutment surface was consistent with the findings of minimum peri-implant sulcus depth and no peri-implant attachment loss. Ferraris et al. have demonstrated that the micromachined surfaces with horizontal grooves 3 or 10 mm deep on implant surfaces interfere with epithelial downgrowth through a contact guidance mechanism [31,32]. Furthermore, Nevins et al. histologically evidenced that connective tissue fibers were perpendicularly oriented to the implant surface and prevent the apical migration of gingival epithelial cells and fibroblasts [11].

The relative position of the gingival margin did not change in relation to the implant crown at all of the three time points, both within and between both groups, indicating that the gingival margin was maintained coronally at all time intervals with a defined and stable biologic width. 

The MCBL at both the mesial and distal aspects immediately after restoration and at 1 year post functional loading in both groups showed no evidence of bone loss. Intragroup evaluation showed a statistically significant minimal crestal bone loss in the mesial and distal aspects of both the control and test groups from baseline to 1 year post functional loading. Mean mesial and distal crestal bone level measurements between the groups at 1 year post functional loading revealed minimal early crestal bone loss with no statistical significance. Both groups had minimal early crestal bone level changes, which was much lesser than Albrekktson’s criteria [33] of permissible early crestal bone loss measurements, with 100% implant survival rate. 

A level of 100% ISR [34,35,36] and similar implant stability were observed in both groups [17,18,19]. The ISQ was greater than the acceptable levels, and thus an immediate loading protocol was followed in both the diabetic and non-diabetic patients. There have been few investigations on immediate functional loading of single-tooth implants. Published results, on the other hand, showed that immediate functional loading of implants with the traditional placement approach and appropriate primary stability could be a viable therapy option. The possibility of rehabilitating the patient’s function and aesthetics in a very short period of time was without doubt attributed to the use of LMG implant surfaces and thus it was proposed that Laser-Lok technology tended to improve hard and soft tissue integration, which might be beneficial to immediate loading. In the present study, an interesting parameter for the immediate loading protocol, the ISQ, was hypothesized and was found to enhance primary stability for immediate functional loading.

Future studies comparing well-controlled, moderately- controlled, and poorly-controlled diabetic patients should be carried out, so as to evaluate if Laser-Lok technology could benefit systemically compromised patients. The small sample size of this study is a limitation and hence further studies based on the present results could be formulated to validate the findings with larger sample sizes involving more prospective parameters.

## 5. Conclusions

To summarize, the study findings indicated that Laser-Lok implants [33] with laser micro-grooved platform switched abutments reduced plaque accumulation 1 year post functional loading in both diabetic and non-diabetic patients, thereby reducing inflammation. The mean crestal bone level changes were very minimal and comparable in both diabetic and non-diabetic patients, in turn maintaining peri-implant sulcus depth and the relative position of the gingival margin. The study suggested that moderately controlled diabetic patients were no longer a contraindication for dental implant survival rate and stability. The additional value of microtexturing on implant and abutment surfaces could be used as an advantage in diabetic individuals to overcome the pathological changes associated with metabolic changes. Furthermore, in an immediate implant loading protocol, laser micro-grooved implants and abutments might mitigate or eliminate the peri-implant mean crestal bone loss in moderately controlled diabetic individuals.

The present study was the first to evaluate clinical and radiographic parameters in laser micro-grooved implants and abutments loaded immediately in moderately controlled diabetic patients. However, in the future, studies comparing well-controlled, moderately-controlled, and poorly-controlled diabetic patients rehabilitated with Laser-Lok implants and abutments could be carried out to take complete advantage of this novel technology. The small sample size was a limitation, and thus further studies with larger sample sizes involving more prospective parameters should be designed.

## Figures and Tables

**Figure 1 medicina-58-01456-f001:**
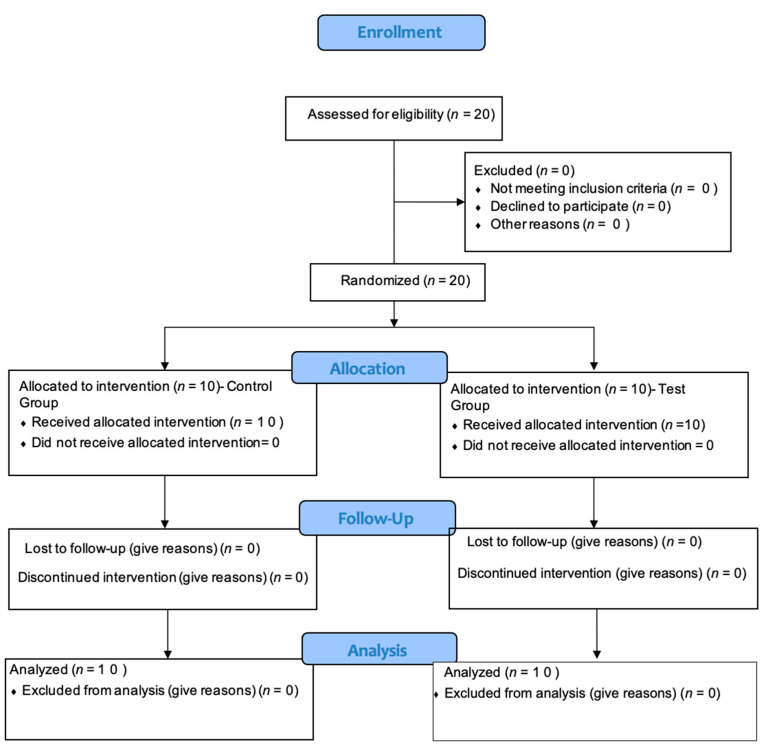
Consort flow diagram.

**Figure 2 medicina-58-01456-f002:**
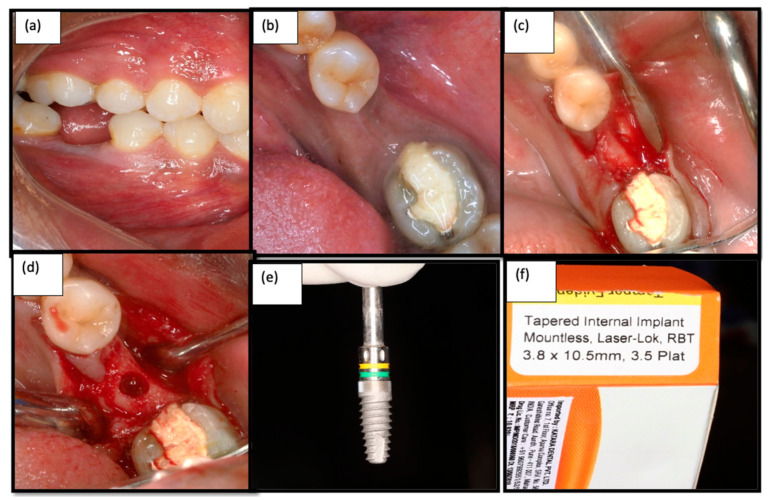
(**a**) Isolated edentulous site—Buccal view; (**b**): Intra oral photograph of the edentulous site—Occlusal view (**c**): Crestal incision given and flap elevation performed (**d**) final osteotomy site preparation done for 3.8 mm; (**e**,**f**) Laser micro-grooved Implants (Laser lok^®^).

**Figure 3 medicina-58-01456-f003:**
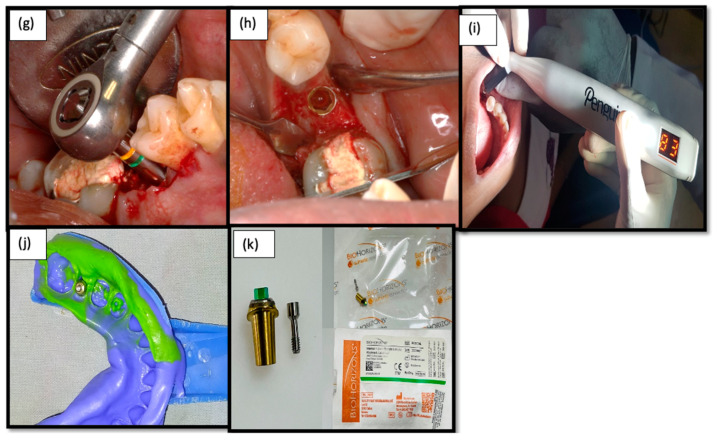
(**g**) Implant driven in using a Bio-horizon torque wrench; (**h**) Laser Micro-grooved implant placed and cover screw given; (**i**) Resonance frequency analysis for prosthetic loading; (**j**) Putty impression taken for prosthetic restoration; (**k**) Laser micro-grooved abutment – Laser Lok^®^.

**Figure 4 medicina-58-01456-f004:**
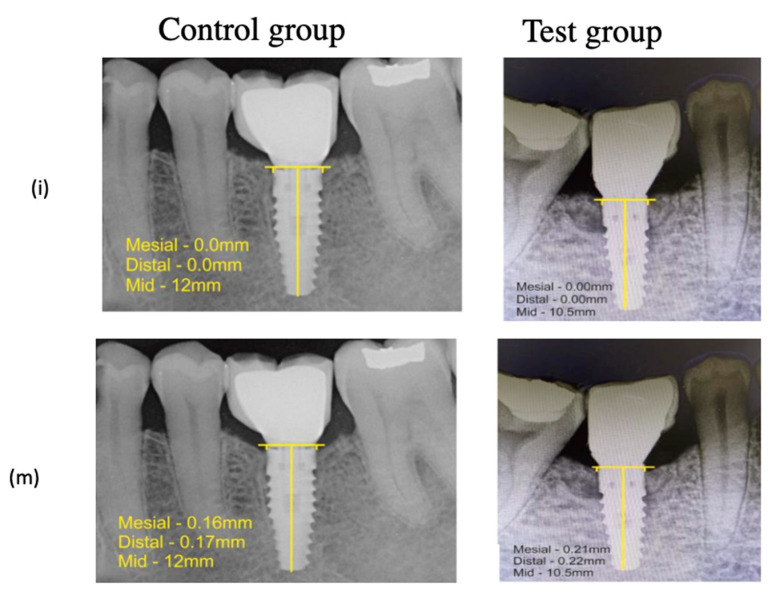
Measurement of Mean Crestal Bone Level—(**i**) Immediately Post functional Loading, (**m**) 1 year Post functional Loading.

**Table 1 medicina-58-01456-t001:** Intergroup and intragroup comparison of clinical parameters at different timepoints.

Clinical Parameter		Control Group (Group 1)	Test Group (Group 2)	*p* Value
PPD	Baseline	2.94 ± 0.16	2.88 ± 0.12	0.43
	1 Year	2.90 ± 0.10	2.63 ± 0.14	0.39
*p* value		0.90	0.66	
CAL	Baseline	0.00	0.00	-
	1 Year	0.00	0.00	-
*p* value		-	-	-
Full mouth plaque scores (FMPS) (%)	Baseline	15.74 ± 2.26	15.15 ± 1.85	0.52
	1 Year	10.62 ± 1.38	10.70 ± 1.30	0.32
*p* value		≤0.001 **	≤0.001 **	
Full mouth bleeding scores (FMBS) (%)	Baseline	11.80 ± 1.20	11.49 ± 1.51	0.51
	1 Year	8.61 ± 0.39	8.53 ± 1.47	0.86
*p* value		≤0.015 *	≤0.015 *	

CAL: Clinical attachment loss; ** highly significant, * significant.

**Table 2 medicina-58-01456-t002:** Intergroup comparison of site-specific clinical parameters at different timepoints.

Clinical Parameter		Control Group (Group 1)	Test Group (Group 2)	*p* Value
S-SPS	Baseline	9.37 ± 6.63	9.37 ± 6.63	1.0
	6 months	8.30 ± 6.70	8.70 ± 6.30	0.90
	1 year	6.25 ± 4.75	6.25 ± 4.75	1.0
S-SBS	Baseline	9.37 ± 6.63	9.37 ± 6.63	1.0
	6 months	8.50 ± 6.50	7.50 ± 5.50	0.66
	1 year	6.37 ± 4.63	6.25 ± 4.75	0.77
PISD	Baseline	2.11 ± 0.58	2.03 ± 0.56	0.80
	6 months	2.09 ± 0.11	2.04 ± 0.13	0.67
	1 year	2.06 ± 0.55	2.07 ± 0.50	0.73
PIAL	Baseline	0.00	0.00	-
	6 months	0.00	0.00	-
	1 year	0.00	0.00	-
R-PGM	Baseline	3.37 ± 0.20	3.53 ± 0.30	≤0.263
	6 months	3.37 ± 0.20	3.53 ± 0.30	≤0.263
	1 year	3.37 ± 0.20	3.53 ± 0.30	≤0.263

S-SPS- Site-Specific Plaque Scores, S-SBS- Site-Specific Bleeding Scores, PISD- Peri-Implant Sulcus. Depth, PIAL- Peri-Implant Clinical Attachment Level, R-PGM- Relative Position of the Gingival Margin.

**Table 3 medicina-58-01456-t003:** Intragroup comparison of clinical parameters between different timelines.

Clinical Parameter	Timeline	Control Group (Group 1) *p* Value	Test Group (Group 2) *p* Value
Site-Specific Plaque Scores (SSPS)%	Baseline to 6 months	0.598	0.69
	Baseline to 1 year	**≤0.05 ****	**≤0.05 ****
	6 Months to 1 year	0.56	0.59
Site-Specific Bleeding Scores (SSBS)%	Baseline to 6 months	0.54	0.59
	Baseline to 1 year	**≤0.05 ***	**≤0.05 ***
	6 Months to 1 year	0.84	0.61
Peri-Implant Sulcus Depth (PISD) (mm)	Baseline to 6 months	0.70	0.89
	Baseline to 1 year	0.89	0.82
	6 Months to 1 year	0.93	0.85
Peri-Implant Clinical Attachment Level (PIAL) (mm)	Baseline to 6 months	-	-
	Baseline to 1 year	-	-
	6 Months to 1 year	-	-
Relative Position of the Gingival Margin (R-PGM) (mm)	Baseline to 6 months	1.0	1.0
	Baseline to 1 year	1.0	1.0
	6 Months to 1 year	1.0	1.0

* significant. ** highly significant.

**Table 4 medicina-58-01456-t004:** Intra- and intergroup comparison of crestal bone loss.

Radiographic Parameter	Timeline	Control Group (Group 1)	Test Group (Group 2)	*p* Value
M-MCBL	Baseline	0.00	0.00	-
	1 Year	0.16 ± 0.15	0.21 ± 0.03	0.40
*p* Value		0.003 *	≤0.001 **	
D-MCBL	Baseline	0.00	0.00	-
	1 Year	0.17 ± 0.16	0.22 ± 0.04	0.48
*p* Value		≤0.001 **	≤0.001 **	

* significant. ** highly significant.

## Data Availability

Not applicable.

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
