# Peer review of "The Effect of Laser Micro Grooved Platform Switched Implants and Abutments on Early Crestal Bone Levels and Peri-Implant Soft Tissues Post 1 Year Loading among Diabetic Patients—A Controlled Clinical Trial"

_medicina, 2022, doi:10.3390/medicina58101456_

Round 1

Reviewer 1 Report

This paper aims to show the effect of laser micro grooved platform switched implants and abutments on healing after 1 year loading through a clinical trial. It is a well structured and organized document.

It will be more clear if only CAD CAM frameworks would be included and comparison based in the material.

Abstract: sections are not identified; soft tissue healing variables are not identified in the objectives and along the abstract but is mentioned in the title. Statistical significance in early CBL reveal differences between control and test groups but in the text says that they are similar. Abbreviations are not identified in the text (ISQ, MCBL- the same abbreviation needs to remain the same along the document. Seems the abstract is incomplete since it does not have a conclusion related to research hypothesis. Results in this abstract are confusing

One of the keywords is not mentioned in the abstract: relative position of the gingival margin- seems like this is the variable related to soft tissues that is missing in the abstract

Introduction must include abbreviations of the main features like Laser Micro-grooved and others (Mean Crestal bone level, , Implant Survival Rate) just after they are mentioned for the first time. Than they can be cited by the abbreviation.

Hypothesis of this research should be mentioned after the objectives are lined.

Material and Methods:

The phrase [Published in the Clinical Oral Implants Research with impact factor with 116 4.305] seems inadequate in the article.

Power analysis to calculate sample size should be mentioned and explained since seems like a little low.

Outcome measures included—seems like variables included are not relevant for the objectives of the article. I suggest to stick to the variables used for this specific study and their explanations.

Inclusion criteria should include local of edentulous arch were the implants were to be placed (different locations on the dental arch could have different occlusal and lateral loads, that can influence osteointegration, which can considered as bias to the results)

This phrase “P≤ 0.05 was considered statistically significant” is a repetition in what was said in the previous sentence. It is not necessary.

Results- variables not relevant for response to research questions or objectives should not be presented nor statistically analysed since they cause confusion

Discussion- discussion should not be another description of results and needs to take in consideration relevant results from previous studies.

Author Response

S no

Reviewer Comments

Corrections done and Reply to Reviewers

1.

 This paper aims to show the effect of laser micro grooved platform switched implants and abutments on healing after 1 year loading through a clinical trial. It is a well structured and organized document.

It will be more clear if only CAD CAM frameworks would be included and comparison based in the material.

Thanks for the suggestion,

The objective of the study was to test the effect of Laser micro-grooved implants and abutments and its ability to maintaining crestal bone levels among Diabetics and Non Diabetics individuals at 1 year post functional loading, and all cases were restored using metal ceramic crowns due to affordability issues.

The suggestion of CAD-CAM will be considered for our future studies as it would open up a new dimension in our research, we would surely consider this in our next study.

2.

Abstract: sections are not identified; soft tissue healing variables are not identified in the objectives and along the abstract but is mentioned in the title. Statistical significance in early CBL reveal differences between control and test groups but in the text says that they are similar. Abbreviations are not identified in the text (ISQ, MCBL- the same abbreviation needs to remain the same along the document. Seems the abstract is incomplete since it does not have a conclusion related to research hypothesis. Results in this abstract are confusing

Structured abstract has been provided.

Title has been slightly changed.

The term Soft tissue healing has been removed and replaced with “peri-implant soft tissues” in the title.

The peri-implant parameters such as Peri-implant attachment level(PIAL) and relative position of the gingival margin(R-PGM) have been included  in the objectives .

The correction regarding Statistical significance has been  corrected and included in the abstract as follows:

“Early MCBL ,within the groups 1 year postloading was similar  both mesially and distally ( Control- 0.00 mm to 0.16 mm and 0.00 mm to 0.17 mm respectively); ( test-0.00 mm to 0.21 mm and 0.00 mm to 0.22 mm respectively)  with  statistical significance [P ≤ 0.003,P ≤0.001] &[P ≤ 0.001,P ≤ 0.001]. However, intergroup comparison showed no significant difference statistically in the MCBL in 1 year post functional loading.The peri-implant soft tissue parameters showed no significant difference between the groups.”

 Results have been modified in the abstract

Abbreviations has been checked and completely corrected in the text 

Conclusion has been corrected in the abstract relevant to research hypothesis

3.

 One of the keywords is not mentioned in the abstract: relative position of the gingival margin- seems like this is the variable related to soft tissues that is missing in the abstract

 The peri-implant soft tissue parameter -Relative position of the gingival margin has been included in the abstract

4.

Introduction must include abbreviations of the main features like Laser Micro-grooved and others (Mean Crestal bone level,Implant Survival Rate) just after they are mentioned for the first time. Then they can be cited by the abbreviation.

Abbreviations have been included for all the main features in the manuscript. Then later on cited in the text by the abbreviation.

5.

 Hypothesis of this research should be mentioned after the objectives are lined.

 Hypothesis of this research has been  mentioned after the objectives

6.

 Material and Methods:

The phrase [Published in the Clinical Oral Implants Research with impact factor with 116 4.305] seems inadequate in the article.

 The requested phrase has been removed

7.

Power analysis to calculate sample size should be mentioned and explained since seems like a little low.

 Sample size calculation was done based on results

obtained from a study by Antonio Aguilar- Salvatierra et al. in 2016, with 1% alpha error and 95% power based on Mean Bone Level measurements with

08 implants in each group with a total of 16 implants.

The following formula were used and sample size calculation was derived using:

Two means – hypothesis testing for two means (equal variances)

Standard Deviation in the 1st group S1 = 0.19

Standard Deviation in the 2nd group S2 = 0.29

Mean difference between 1st and 2nd sample = 0.1

Effect size = 2.6100666

Alpha Error (%) = 1

Power (%) = 95

Sided = 2

Sample Size = 16 (08 in each group)

Hence keeping in mind that 20% could be dropouts we had considered 10 implants per group.

All patients recruited completed the study without any dropouts.

8.

Outcome measures included—seems like variables included are not relevant for the objectives of the article. I suggest to stick to the variables used for this specific study and their explanations.

The reasons for including variables such as

FMBS & FMPS : as less than 20% were only considered for surgical procedure

S-SPS & S-SBS : these variables were assessed so that they represent the amount of plaque and bleeding present in the site specific implant sites so that they can be correlated to patients' maintenance and inflammation which could influence the outcome ( Mean crestal Bone level changes).

PPD & CAL & Position of Gingival Margin : these variable give us the overall outcome of the soft tissue representation which could play a significant role in bone level maintenance. Position of Gingival Margin is one of the secondary objectives of the study.

All the above mentioned variables would be beneficial which could correlate with the primary outcome.

9.

Inclusion criteria should include local of edentulous arch were the implants were to be placed (different locations on the dental arch could have different occlusal and lateral loads, that can influence osteointegration, which can considered as bias to the results)

As mentioned by the reviewers we agree that  differing locations would cause differing loads; therefore it was already planned in the study design and the study recruited only Mandibular premolar and molar sites( inclusion criteria) and it was evenly distributed among both the groups so that no bias could be appreciated for the same.

10.

 This phrase “P≤ 0.05 was considered statistically significant” is a repetition in what was said in the previous sentence. It is not necessary.

 This phrase has been deleted

11.

Results- variables not relevant for response to research questions or objectives should not be presented nor statistically analysed since they cause confusion

 The reasons for including variables such as

FMBS & FMPS : as less than 20% were only considered for surgical procedure

S-SPS & S-SBS : these variables were assessed so that they represent the amount of plaque and bleeding present in the site specific implant sites so that they can be correlated to patients' maintenance and inflammation which could influence the outcome ( Mean crestal Bone level changes).

PPD & CAL & Position of Gingival Margin : these variable give us the overall outcome of the soft tissue representation which could play a significant role in bone level maintenance. Position of Gingival Margin is one of the secondary objectives of the study.

All the above mentioned variables would be beneficial which could be correlated with the primary outcome.

12.

Discussion- discussion should not be another description of results and needs to take in consideration relevant results from previous studies.

Discussion has been modified  accordingly

REVIEWER 2 COMMENTS AND CLARIFICATIONS:

S no

Reviewer Comments

Corrections done and Reply to Reviewers

1.

Dear colleagues, thank you for the opportunity to review your article.

I read it with interest, but it raises a number of questions that should be said.

Introduction

1. In general, the treatment of comorbid patients is our common concern due to diagnostic and treatment options, as well as the individual response of the patient's body. You know that the covid-19 pandemic has taken a toll on health, especially for patients with diabetes.

this section needs serious revision: the list of references refers to studies that are more than 10, 20, and even 30 years old, and the features of the last few are practically not reflected.

All patients who were recruited for the study underwent RT-PCR for Covid -19 and those patients who turned negative were only recruited for the implant placement.

List of references  has been modified

Materials and Methods

1. What type of diabetes was the focus of the study? And what criteria of this disease, regarding bone tissue, played the role of inclusion, non-inclusion and exclusion. Were HbA1c levels of 8.1 to 10 sufficient for the assessment?

It has been established that the sixth complication of diabetes is periodontitis in which  bone loss is one of the hallmark signs.

The study objective is to assess moderately controlled diabetes and hence patients with  HbA1c levels of 8.1 to 10 were recruited.

2. Why this implant system was used - Biohorizon's Implant. If the choice was empirical, then the commercial name can be excluded, if the study was targeted at a specific type of implant, it should also be written about and clarified about the conflict of interest.

 The commercial name Biohorizon's Implant has been removed wherever it was mentioned.

Results

In general, according to the study, I consider it correct to replace men -> male and women -> women.

Were the authors noticed complications or adverse events in terms of significant differences?

The words men were replaced with male.

No complications or adverse events were observed in the treated patients.

I want to draw attention to the quality of an artificial crown as a discussion and features of the rehabilitation of patients with diabetes. It is necessary to restore the equator of the tooth more carefully and avoid planar options for the contact point. In this regard, in the test group in figure 4, you need to comment on the state of the bone tissue and its loss.

All patients were checked for occlusal evaluation and the mesial and distal contacts with the adjacent tooth were thoroughly elicited using a dental floss. After establishing the contacts screw retained crowns were restored permanently. All patients came in for 6 months and 1 year post functional evaluation - the restored crowns did not have any screw or crown loosening - this indicates that all crowns were stable and functional throughout the course of the study.

Radiographs in figure 4 show slight changes in crestal bone levels (0.22) 1 year post functional loading which are very minimal and negligible than that of the accepted Albrektsson criteria of 1-1.5mm post 1 year of functional loading

The discussion also needs to be supplemented, taking into account the list of references (as well as in the introduction).

The discussion has been  supplemented according to the given references and highlighted.

Reviewer 3: COMMENTS AND CLARIFICATIONS:

S no

Reviewer Comments

Corrections done and Reply to Reviewers

Add more references (2021-2022). You can add [1] New Titanium Alloys, Promising Materials for Medical Devices; [2] Cytocompatibility of pure metals and experimental binary titanium alloys for implant materials;

Supplementary material related to newer titanium alloys, cytocompatibility and micro-structural modifications have been added with references in the discussion section.

1.Baltatu MS, Vizureanu P, Sandu AV, Florido-Suarez N, Saceleanu MV, Mirza-Rosca JC. New titanium alloys, promising materials for medical devices. Materials. 2021 Oct 9;14(20):5934.

2.Sarraf M, Rezvani Ghomi E, Alipour S, Ramakrishna S, Liana Sukiman N. A state-of-the-art review of the fabrication and characteristics of titanium and its alloys for biomedical applications. Bio-design and Manufacturing. 2021 Oct 26:1-25.

3.Park YJ, Song YH, An JH, Song HJ, Anusavice KJ. Cytocompatibility of pure metals and experimental binary titanium alloys for implant materials. Journal of dentistry. 2013 Dec 1;41(12):1251-8.

4.Liu, Xiaotian, et al. "Binary titanium alloys as dental implant materials—a review." Regenerative biomaterials 4.5 (2017): 315-323.

Show the novelty of the paper compared to the literature, however there is still much work on this topic.

The use of Laser micro grooved implants and abutments have reduced the mean crestal bone loss in both Diabetic and Non-Diabetic groups and also the micro-grooved in the abutments have enhanced the soft tissue attachment towards the abutment surface, which have created a soft tissue cuff and enhanced the Biologic width and thus enhanced soft and hard tissue parameters.

 Thus the novelty of the study is the combined use of Laser micro-grooved implants and abutments have minimized mean crestal bone loss and enhanced  soft tissue parameters in both Diabetic and Non Diabetic patients.

  In the Introduction section, the last paragraph should contain the scientific contribution and scientific hypotheses of your research. Complete, further elaborate the scientific contribution and scientific hypotheses of your research. Be explicit. In addition to the goal of the research (which was written), the novelty in the context of the scientific contribution should be pointed out. Scientific contributions should be written based on the shortcomings of previous research in the literature. In this way, the authors will better emphasize novelty and scientific soundness.

Scientific hypothesis has been mentioned in the introduction

The scientific  contribution in this study reveals that patients with moderately controlled  diabetic patients can also receive dental implants and not only have 100% success rate but also minimize or equate mean crestal bone loss in comparison to Non Diabetic individuals.

Shortcomings of the previous studies have been cited in the introduction section.

Complete the conclusions with the limitations of the proposed methodology. Also write future research.

Limitations and future research has been included in the last para of discussion

Generally, the quality of the writing could be improved.

Sugesstion taken and addressed

Reviewer 2 Report

Dear colleagues, thank you for the opportunity to review your article.

I read it with interest, but it raises a number of questions that should be said.

Introduction

1. In general, the treatment of comorbid patients is our common concern due to diagnostic and treatment options, as well as the individual response of the patient's body. You know that the covid-19 pandemic has taken a toll on health, especially for patients with diabetes.

this section needs serious revision: the list of references refers to studies that are more than 10, 20, and even 30 years old, and the features of the last few are practically not reflected.

Materials and Methods

1. What type of diabetes was the focus of the study? And what criteria of this disease, regarding bone tissue, played the role of inclusion, non-inclusion and exclusion. Were HbA1c levels of 8.1 to 10 sufficient for the assessment?

2. Why this implant system was used - Biohorizon's Implant. If the choice was empirical, then the commercial name can be excluded, if the study was targeted at a specific type of implant, it should also be written about and clarified about the conflict of interest.

Results

In general, according to the study, I consider it correct to replace men -> male and women -> women.

Were the authors noticed complications or adverse events in terms of significant differences?

I want to draw attention to the quality of an artificial crown as a discussion and features of the rehabilitation of patients with diabetes. It is necessary to restore the equator of the tooth more carefully and avoid planar options for the contact point. In this regard, in the test group in figure 4, you need to comment on the state of the bone tissue and its loss.

The discussion also needs to be supplemented, taking into account the list of references (as well as in the introduction).

Author Response

(The authors gave the same response as above.)

Reviewer 3 Report

·       Add more references (2021-2022). You can add [1] New Titanium Alloys, Promising Materials for Medical Devices; [2] Cytocompatibility of pure metals and experimental binary titanium alloys for implant materials;

·        Show the novelty of the paper compared to the literature, however there is still much work on this topic.

·       In the Introduction section, the last paragraph should contain the scientific contribution and scientific hypotheses of your research. Complete, further elaborate the scientific contribution and scientific hypotheses of your research. Be explicit. In addition to the goal of the research (which was written), the novelty in the context of the scientific contribution should be pointed out. Scientific contributions should be written based on the shortcomings of previous research in the literature. In this way, the authors will better emphasize novelty and scientific soundness.

·       Enunciate with a hyphen all Clinical parameters.

·       Complete the conclusions with the limitations of the proposed methodology. Also write future research.

·       Generally, the quality of the writing could be improved.

Author Response

(The authors gave the same response as above.)

Round 2

Reviewer 2 Report

Hello dear authors!

Thank you for your comments. I am satisfied with your work

Reviewer 3 Report

The paper was improved.